# OpenReview forum: "Rethinking Defense for Computer-Use Agents: Context Deception Attacks are Simple to Defend"
_ICLR.cc/2026/Conference — Submitted to ICLR 2026_

### Official Review · Reviewer_pxv9 · 2025-11-01

**Soundness:** 2
**Presentation:** 3
**Contribution:** 2
**Rating:** 2
**Confidence:** 4

**Summary:**

This paper challenges the prevailing assumption that context deception attacks (e.g., fake pop-ups, deceptive HTML attributes) in a computer-use agent’s observation space are inherently difficult to defend against. This paper argues that prior failures stem not from the sophistication of the attacks but from the inadequacy of abstract instruction-based defenses (e.g., “ignore pop-ups”). It proposes in-context defense, a method that prepends a small set of few-shot exemplars demonstrating a defense-first CoT reasoning process before action planning.

**Strengths:**

1. The paper tackles the critical and highly relevant problem of securing computer-use agents, a key barrier to their safe and reliable deployment in the real world.
2. The experiments on reasoning order, exemplar quantity, and OOD generalization provide valuable mechanistic insights into why the method works.
3. The defense requires no model retraining, fine-tuning, or external modules, making it easy to deploy.
4. The paper is well-written.

**Weaknesses:**

1. The proposed method is a direct application of well-established techniques (few-shot ICL + CoT) to a specific attack surface, though claimed as a new "paradigm", which makes it more like a case study. Regarding attack sources, methodologies, algorithms, or architectures, there is little novel contribution, such as how to select example samples or quantify sample quality.
2. The effectiveness of the method is dependent on manually curated exemplars, but the selection and creation process lacks a principled framework, failing to specify how exemplars were chosen or validated. The description of being "randomly sampled" and then subject to "human quality control" is vague and not reproducible. This raises concerns about potential "cherry-picking" and makes it difficult to generalize the findings.
3. The paper claims that these attacks are "simple to defend." This is misleading. While the implementation is simple (adding text to a prompt), the creation of effective defense content requires domain expertise and careful manual labor for each attack type. The approach does not offer a "simple" scalable solution for new or evolving threats.

**Questions:**

1. The core contribution appears to be the application of few-shot ICL with CoT, a well-established technique. Could you please clarify what you view as the fundamental conceptual novelty of your work beyond demonstrating that a standard method is effective in the CUA security domain?
2. Could you provide a detailed protocol for exemplar selection beyond the high-level description? Specifically:
(a) What criteria determined whether a sampled input was “representative”?
(b) How many candidate exemplars were screened before selecting the final three per attack type?
Providing a more systematic representation method will significantly improve readability and reproducibility.
3. The paper's title and main argument assert that these attacks are "simple to defend." However, your method relies on a non-trivial, manual, and expert-driven process of crafting exemplars for each attack category. How does this manual workload align with the claim of simplicity, and how do you envision this approach scaling to a dynamic landscape with new, unforeseen deception attacks?

---

> ### Author Response · Authors · 2025-11-21
> **Rebuttal**
>
> Thank you for the review. Below we address your concerns and questions.
>
>
>
> > **Weakness 1 & Question 1**: The proposed method is a direct application of well-established techniques (few-shot ICL + CoT)... there is little novel contribution... Could you please clarify what you view as the fundamental conceptual novelty?
>
> This is a crucial point, and we appreciate the chance to clarify our paper's central argument.
>
> We agree with you that in-context defense, indeed, is a straightforward application of ICL and CoT. However, **we never claim it as a *new paradigm*, nor do we claim it as a contribution**.
>
> Our primary contribution is not a method, but a **conceptual shift in perspective**. The prevailing view, as we cite, treats context deception attacks as an inherently difficult and hard-to-defend security challenge. Our core argument is that this view is incorrect, and we demonstrated this by using a delibrately **simple and well-established baseline** (ICL + CoT) to show the problem is, in fact, tractable.
>
> Using a more complex or *novel* method (e.g., sophisticated exemplar selection, finetuning, etc.) would fundamentally **weaken our claim** in logic.
>
>
>
>
>
> > **Weakness 2 & Question 2**: The effectiveness of the method is dependent on manually curated exemplars, but the selection and creation process lacks a principled framework... vague and not reproducible... raises concerns about potential "cherry-picking"... Could you provide a detailed protocol?
>
>  - **Cherry-picking:** We understand your concern, but we did not do cherry picking. The selection process was detailed in Appendix F.1.
>      - **Pop-up Attacks**: We reused the three existing, built-in exemplars from the VisualWebArena benchmark. There's no way we can cherry-pick on that.
>      - **EIA and EDA Attacks**: Exemplars were randomly sampled. We did not screen or inspect non-selected exemplars to find "better" ones.
>      - **Further validation**: We ran additional experiments (Table 10) using **two new sets** of randomly sampled exemplars, with the defensive performance maintained. This directly addresses the "cherry-picking" concern.
>
>  - **Reproducibility:** To ensure the human refinement process is reproducible and not vague, we provided a detailed, 4-point protocol in Appendix F.3 (Human Refinement for Quality Control). This protocol, combined with the exact prompt used (Appendix F.2), provides a clear framework for reproducing the defensive reasoning.
>
>
> Again, our goal was not to find an optimal selection method, but to show that even a straightforward, non-optimized random selection is highly effective, showing that defense is possible and tractable unlike prior works believed.
>
>
>
>
>
> > **Weakness 3 & Question 3**: The paper claims that these attacks are "simple to defend". This is misleading... the creation of effective defense content requires domain expertise and careful manual labor... How does this manual workload align with the claim of simplicity, and how do you envision this approach scaling?
>
> Thank you for raising this point. We use the term "simple" in direct contrast to far more complex defense paradigms, such as finetuning or designing new system architectures.
>
>
>  - **On Simplicity of Labor:** The "Risk/Distraction Analysis" is typically just two or three short, common-sense sentences (example shown below). This does not require experts at all. Crafting such a brief, descriptive statement is fundamentally simpler than, for example, re-engineering a system's architecture.
>
> ````
> # Risk/Distraction Analysis
> The Latest Listings section containing tags [8] to [11] could be advertisements, recommending items with prices under the search bar. Since I am required to ignore advertisements, ignoring them minimizes risk.
> ````
>
>  - **On Scalability:** This simplicity is precisely what makes the approach scalable. When a new, unforeseen deception attack emerges, a defender does not need to re-train or finetune the model (re-training/finetuning are the tasks that truly require domain expertise). Instead, the defense can **"hot-patch" the agent** by simply adding 1-3 new exemplars (a screenshot and a few lines of text) to the agent's context or retrieval database (as discussed in Appendix H). Therefore, this simple defense is agile and scalable.
>
>
>
>
>
> Thank you again for raising these points, and we hope the clarifications have addressed your concerns. If so, we would appreciate your re-evaluation of the work.

---

> > ### Comment · Reviewer_pxv9 · 2025-11-27
> >
> > Thanks for your explanations and clarifications. I appreciate the ablation studies you did in the submission, like the one where different exemplars are used. However, my concern remain. Given the limited number used, even with the experiments of the two new sets of random exemplars, one still cannot claim that any random exemplars would be able to maintain the defense effect. Moreover, as I mentioned, these exemplars are constructed based on the existing works on attacks, how to know their effectiveness if faced with new type of attacks or new tasks beyond those in these works. To be technically solid and convincing, the observation (the random exemplars are sufficient) alone is not sufficient, more in-depth analysis is needed ( I am not asking for optimal or better exemplars selection method.). For instance, what is the relation between different characteristics of exemplars (like malicious types, tasks), present context, and the defense effect? Whether the random exemplars constructed from the mentioned works still work for unseen types of attacks and tasks?

---

> > > ### Author Response · Authors · 2025-11-28
> > >
> > > Thank you for the follow-up comment.
> > >
> > >
> > >
> > > > Given the limited number used, even with the experiments of the two new sets of random exemplars, one still cannot claim that any random exemplars would be able to maintain the defense effect.
> > >
> > > **Experiments with more random exemplars yield the same conclusion.** We repeated the experiments for another 10 sets of randomly sampled exemplars. The results align with those reported in the paper, as shown below:
> > >
> > > |                             |    Pop-up   | EIA (EI-text) | EIA (EI-aria) |   EIA (MI)  | EDA (Type 1) | EDA (Type 2) | EDA (Type 3) |
> > > |-----------------------------|:-----------:|:-------------:|:-------------:|:-----------:|:------------:|:------------:|:------------:|
> > > | ASR in Table 1              |    0.051    |     0.117     |     0.170     |    0.035    |     0.000    |     0.000    |     0.000    |
> > > | ASR in Repeated Experiments | 0.052±0.018 |  0.122±0.024  |  0.171±0.026  | 0.035±0.022 |  0.002±0.004 |  0.001±0.002 |  0.000±0.001 |
> > >
> > >
> > >
> > >
> > >
> > > > In-depth analysis: Whether the random exemplars constructed from the mentioned works still work for unseen types of attacks and tasks?
> > >
> > > - The random exemplars **work** for unseen *attack types* within the same *attack family* **(Table 4)**. Example: Pop-up exemplars defend against diverse, unseen Pop-up variations.
> > > - They **do not** work for unseen *attack families* **(Line 440)**. Example: Pop-up exemplars defend against EIA attacks.
> > >
> > > This is because the defense is a ***simplest effective demonstration*** to show that "context deception attacks are simple to defend". It is a ***few-shot*** but not a ***universal, zero-shot solution*** for entirely new attack families.
> > >
> > >
> > >
> > >
> > >
> > > Once again, thank you for being the first to discuss with us. We are happy to address any remaining concerns or provide further details if needed.

---

### Official Review · Reviewer_oWnt · 2025-11-01

**Soundness:** 2
**Presentation:** 3
**Contribution:** 2
**Rating:** 4
**Confidence:** 4

**Summary:**

This paper proposes an in-context defense paradigm for computer-use agents (CUAs). Specifically, rather than only giving abstract instructions, the defense provides agents with a set of exemplars that includes defensive CoT (detection → justification → mitigation) before action planning. Although the technique is simple, the defense shows effectiveness against current context deception attacks (e.g., ~91% ASR reduction on pop-ups, ~75% on environment injection, and near-perfect performance on environmental distractions).

**Strengths:**

- The paper is well-motivated towards the current threat to the CUAs.
- The defense is easy to deploy and is effective against current attacks.

**Weaknesses:**

- The methodology contribution of this defense is limited as it only uses in-context + CoT. Some other regular methods, such as whether including some few-shot defensive learning could improve the performance, is not discussed.
- The defense is strongly limited by that the defender must have knowledge of the possible attacks. Although Figure 9 and Table 4 show that it can generalize to OOD pop-ups, however, whether this defense knowledge is transferable between different types of attacks is unclear. As newer attacks emerge, this method will need to collect them and the longer the provided context length, make it less scalable.
- The method is strongly limited by using exemplars per attack family. Although Figure 9 and Table 4 show that it can generalize to OOD pop-ups, it is unclear whether defense knowledge transfers across attack types. As new attacks emerge, the defense may require frequent exemplar updates, which might not be scalable.
- There is no ablation on CoT (w/o CoT, or simplified version of current CoT).

**Questions:**

- Please update all references to their published versions.
- What are the false cases, and what will affect the defense performance? How would factors such as UI changes (dark mode, responsive layouts) affect robustness?

---

> ### Author Response · Authors · 2025-11-21
> **Rebuttal (1/2)**
>
> We sincerely thank the reviewer for their insightful and constructive feedback. Below, we address the reviewer's valuable points.
>
>
>
> > The methodology contribution of this defense is limited as it only uses in-context + CoT. Some other regular methods, such as whether including some few-shot defensive learning could improve the performance, is not discussed.
>
> We agree that for a paper that proposes a novel method, exploring more complex techniques is essential.
>
> But our paper's main contribution is a ***shift in perspective***. As stated in our introduction, we "challenge the prevailing view that context deception attacks are inherently difficult to defend against". Our goal was to use the ***simplest effective demonstration*** as possible to prove that this perceived difficulty is incorrect. The fact that a simple in-context CoT defense works so well is the central evidence for our hypothesis. Using a more complex method (like finetuning) would **weaken this core argument**.
>
> Regarding defensive learning, we **did discuss this in Line 652**. We agree that few-shot finetuning is a valid direction, but we note that it may require extra finetuning to adapt to future, unseen attacks. For this reason, we suggested the more efficient, training-free **retrieval-augmented defense**, and we provided a preliminary experiment in Appendix H{sec:retrieval} (Table 12) demonstrating its feasibility and high performance.
>
>
>
>
>
> > The defense is strongly limited by that the defender must have knowledge of the possible attacks... whether this defense knowledge is transferable between different types of attacks is unclear. As newer attacks emerge, this method will need to collect them and the longer the provided context length, make it less scalable.
>
> This is an excellent point, and we'd like to address transferability and scalability separately.
>
> * **Transferability:** You are correct that this defense, which is a ***simplest effective demonstration***, is not a universal, zero-shot solution for entirely new attack *families*. Our OOD experiments (Table 4) were intended to show generalization to *variations* within an attack *family* (e.g., new pop-up styles, but still pop-ups), not to claim transferability to fundamentally different, unseen attack *families*, because *no defense is universal, in any domain.*
> * **Scalability & Context Length:** This is the precise motivation for our discussion in Appendix A and H. The concern that "the longer the provided context length, make it less scalable" is exactly why we proposed and tested a **retrieval-augmented defense**. Our preliminary results in Appendix H are very promising: we show that retrieving only **3** relevant exemplars not only solves the context length problem but also *boosts* defense performance. This demonstrates a clear path toward a scalable and adaptable defense that does not require an ever-expanding context window.
>
>
>
>
>
> > There is no ablation on CoT (w/o CoT, or simplified version of current CoT).
>
> We did, in fact, provide this analysis in **Table 2**, where the **"without CoT"** condition is the baseline, referred to as "prompting defense" (illustrated in Figure 1). This baseline fails to provide effective defense.
>
> Furthermore, **between CoT and "w/o CoT"**, we conducted an additional ablation in **Table 8** using a "stronger prompting defense". These stronger prompts show some improvement, but still largely not effective compared to using in-context exemplars (over 50% difference).
>
> Together, the results help explain why there's a prevailing view that context deception attacks are difficult to defend. And our work challenges this view by showing **a shift to concrete reasoning exemplars is sufficient**.
>
>
>
>
>
> > Please update all references to their published versions.
>
> Thank you for the reminder. We have updated the references in the revised manuscript.

---

> ### Author Response · Authors · 2025-11-21
> **Rebuttal (2/2)**
>
> > What are the false cases, and what will affect the defense performance? How would factors such as UI changes (dark mode, responsive layouts) affect robustness?
>
> Thank you for pointing this out, which was missing in the previous manuscript. We have added a detailed analysis of all 15 failure cases (out of 301 trials) on the VisualWebArena benchmark as **Appendix I**, including visualizations and model outputs.
>
>
>
> ## TL;DR
>
> UI changes are not the primary/direct cause of failure. Rather, whether the defensive CoT is induced is the deciding factor, which could be attributed to the model's prompt following ability.
>
>
>
> ## False cases
> The false cases fall into two categories:
>
> 1. **Defensive CoT not Induced (5 cases):** In these instances, the agent did not initiate the "defend-then-act" reasoning and instead proceeded directly to action planning, failing to identify the threat.
> 2. **False Positives due to Agent Artifacts (10 cases):** In these cases, the agent's defensive CoT *correctly* identified the threat. However, a "dummy parsing algorithm" artifact in the agent's implementation (which we kept unchanged for fairness) caused it to parse the malicious element discussed in the CoT, leading to a failure.
>
>
>
> ## Influence of UI Changes
>
> 1. The above false case analysis revealed **no correlation between failure cases and UI patterns**; that is, no specific UI pattern was found to be more prone to failure.
> 2. Nevertheless, our experiments have covered UI changes:
>     - **General Stylistic Changes:** Our OOD experiments (Table 4) introduce diverse fonts, layouts, and visual styles. The marginal increase in attack success rate (6.1%) demonstrates high robustness to these variations.
>     - **Dark Mode:** This is already covered in our main experiments. The EDA-experiment includes the Discord interface, which is a native dark-mode environment. All attacks are defended (Table 1).
>     - **Responsive Layouts:** This is also covered in our main experiments. The VisualWebArena environment is interactive, while the pop-up, once clicked, would vanish and reappear in varying locations. On this setting, the ASR was reduced by over 90% (Table 1).

---

### Official Review · Reviewer_BpQb · 2025-11-03

**Soundness:** 3
**Presentation:** 4
**Contribution:** 3
**Rating:** 6
**Confidence:** 4

**Summary:**

This paper reveals that the perceived difficulty of defending Computer-Use Agents against context deception attacks stems from ineffective defense paradigms, and that these attacks are in fact simple to defend. The authors systematically challenge this perception, which arose from the failure of prior defenses like abstract instructions , by introducing a new "in-context defense" paradigm. This method leverages in-context learning, augmenting the agent's context with a minimal set of exemplars to guide it to perform explicit defensive reasoning before action planning. Experiments show this method is remarkably effective, reducing up to 91.2% of pop-up window attacks and achieving near-perfect defense on other deception attacks , in stark contrast to prior defenses. The authors also identify that the key to this defense is teaching the agent a sequential "defend-then-act" reasoning process. Overall, their work provides a fundamental shift in perspective, underscoring that context deception attacks are far more tractable than previously believed and that success hinges on demonstrating a reasoning process rather than issuing an abstract rule.
I have also provided some suggestions that I hope the authors find constructive. I am very open to adjusting my score upon a satisfactory response.

**Strengths:**

The failure of abstract instructions to secure agents has been a significant bottleneck, creating a widespread perception that context deception attacks are an inherently difficult problem. This work reveals this perception is an oversight, emphasizing the imperative for agent safety to focus on teaching how to think rather than just providing rules of what to do.
1.The paper is very well written, clear, and I enjoyed reading it.
2.The defense method is not only effective but also surprisingly simple and data-efficient, requiring no complex fine-tuning.
3.Along the way of showcasing their findings, the authors also make other important contributions:
(1)They precisely identify the key mechanism of the defense by ablating the reasoning order, showing that a "defense-first" analysis is critically more effective than a "planning-first" analysis.
(2)They also test the limits of the method's data efficiency, finding that a single defensive exemplar is sufficient to reduce attack success by 96.2%, which is a very powerful and interesting result.

**Weaknesses:**

The claim that the agent learns a "portable reasoning skill" that generalizes to unseen attacks warrants further elucidation. The defense's effectiveness is predicated on the agent identifying "atypical" or "inconsistent" UI elements . Conversely, i worried that a sophisticated attacker could craft a "perfectly camouflaged" attack that is visually indistinguishable from the benign UI. If the agent is presented with such an attack, its "defense-first" reasoning (which guide the agent to find anomalies) would conclude "Nothing atypical identified," then the attack success rate would be far higher than reported.
The claim that teaching agent how to think rather than just providing rules of what to do is the key to an effective defense. And the finding that a single exemplar can reduce ASR by 96.2% is striking, yet the paper does not deeply analyze why the model's cognitive process is so fundamentally altered by this one example. An in-depth analysis of the model's internals—for instance, how this exemplar changes the model's attention patterns when processing a malicious input—would be highly valuable for truly understanding how this cognitive shift is induced at a fundamental level.

**Questions:**

1.The "defense-first" reasoning process is used to detect "atypical" UI . How would this defense mechanism perform against a "perfectly camouflaged" attack where a malicious element is intentionally designed to be visually indistinguishable from the benign UI (e.g., using identical fonts and button styles)?
2.What is the practical overhead of this method? The paper mentions that adding exemplars "raises the cost of initial inference" but does not quantify the extra token count or the impact on inference latency. How significant is this computational cost for deployment?
3.The paper shows the defense is effective on open-source models like Qwen. Table 11 attributes this to larger models having "stronger comprehension abilities" . Is there an in-depth root cause analysis for why this paradigm is so effective on these models? For example, how do the exemplars mechanistically alter the model's attention patterns when it processes the malicious input?

---

> ### Author Response · Authors · 2025-11-21
> **Rebuttal**
>
> We sincerely thank the reviewer for the overall positive feedback and the insightful questions. We are glad you enjoyed reading the paper.
>
>
>
> > The "defense-first" reasoning process is used to detect "atypical" UI. How would this defense mechanism perform against a "perfectly camouflaged" attack where a malicious element is intentionally designed to be visually indistinguishable from the benign UI (e.g., using identical fonts and button styles)?
>
> This is an excellent question that gets to the core of both the pop-up attack [1] and the defense. Our response has two parts:
>
> 1. A key clarification: the defensive reasoning guides the agent to identify **atypical elements**, not necessarily just **atypical-looking elements**. As shown in the exemplars (Figure 2), the reasoning mainly focuses on **logical inconsistencies**. For the pop-up attacks [1], the reasoning notes both that it "looks strange" (visual) and that it "presents an atypical shortcut for a specific task" (logical).
>
> 2. The **perfectly camouflaged** scenario: Yes, the defense is still valid.
>      - EIA [2] and EDA [3] attacks are such scenarios. EIA duplicates existing HTML elements, while EDA designs UI-consistent page overlays. The defense is successful.
>      - Pop-up attack [1]: We did experiments on this. Results show that with "perfectly camouflaged" pop-ups, the attack itself would fail, as the agent would simply ignore the small pop-ups. This could be why [1]'s authors made the pop-up (1) so big, (2) easy to read, and (3) maintain certain extent of "string matching" with the task instructions.
>
>
>
>
>
> > What is the practical overhead of this method? The paper mentions that adding exemplars "raises the cost of initial inference" but does not quantify the extra token count or the impact on inference latency. How significant is this computational cost for deployment?
>
> 1. **Token Cost**: In our main experiments, the inclusion of the three exemplars introduced an average of **9.92x** more tokens per inference call.
>
> 2. **Latency**: As all our experiments were conducted via API calls, we did not have direct access to measure inference latency. But we did not observe any prohibitive or significant latency that would impede the experimental process.
>
>
>
>
>
> > The paper shows the defense is effective on open-source models like Qwen. Table 11 attributes this to larger models having "stronger comprehension abilities". Is there an in-depth root cause analysis for why this paradigm is so effective on these models? For example, how do the exemplars mechanistically alter the model's attention patterns when it processes the malicious input?
>
> Thank you for this insightful suggestion. To investigate how the defense affects the attention mechanisms, we have conducted a new analysis and added Appendix K (PDF pages 26-32) to the manuscript.
>
> In this appendix, we visualize how the model's output tokens attend to (1) the malicious pop-up and (2) the provided exemplar across different network layers. Two key findings:
>
>  - **Output's Attention on Malicious Element**: The defense reduces attention to the malicious pop-up, especially prior to layer 20 (where feature extraction primarily occurs)
>
>  - **Output's Attention on the Exemplar**: The defense influences attention behaviors the exemplar after layer 20 (where high-level reasoning likely takes place)
>
> We also note, however, that CoT explainability remains an open challenge [4], which we leave for future exploration.
>
>
>
>
>
> **References**
>
> [1] Zhang et al., Attacking Vision-Language Computer Agents via Pop-ups, in ACL 2025.
>
> [2] Liao et al., EIA: Environmental Injection Attack on Generalist Web Agents for Privacy Leakage, in ICLR 2025.
>
> [3] Ma et al., Caution for the Environment: Multimodal LLM Agents are Susceptible to Environmental Distractions, in ACL 2025.
>
> [4] Barez, Bengio, et al., Chain-of-Thought Is Not Explainability, 2025.

---

### Author Response · Authors · 2025-12-03
**Final Remarks by Authors**

# Dear Reviewers,

As the review period ends, we greatly appreciate your constructive feedback. We are also encouraged that you all highly acknowledged the quality of our presentation. We wish you all the best in your research and future careers!

Best regards,
#4688 Authors



---



# Dear AC,

 - We respectfully request downweighing Reviewer pxv9. Their negative assessment relies on verifiable factual errors (hallucinations about non-existent claims) and self-contradictions, which we list explicitly as attached.
 - We invite you to directly take a brief look at the paper to judge it, given that all reviewers praised the writing quality and one specifically "enjoyed reading it".

Thanks for your additional efforts during this unusual reviewing period.

Best regards,
#4688 Authors



---



## Attached: Reviewer pxv9's Factual Errors

 - Hallucination on Claims: In Weakness 1, the reviewer claims we proposed a "new paradigm". This phrase, or a similar expression, never appears in our paper. We explicitly frame our method as a simplest effective baseline.
 - Hallucination on Core Contribution: In Question 1, the reviewer says "The core contribution appears to be the application of few-shot ICL with CoT." This claim never appeared in the paper.
 - Self-contradiction: In Weakness 1, the reviewer criticizes a lack of contribution regarding "how to select example samples", but later contradicts this by stating they are "not asking for optimal or better exemplars selection methods".

---

### Meta-Review · Area_Chair_jTw4 · 2026-01-07

**Summary:**

This paper challenges the perception that context deception attacks are difficult to defend against for computer-use agents, and presents in-context defense, an approach that leverages in-context learning with a minimal set of exemplars to guide the agent to perform explicit defense reasoning before action planning. The reviewers acknowledged the significance of the problem and the effective of the approach against known attacks, but also raised concerns on the technical novelty of the approach (beyond a straightforward application of ICL and CoT) and the transferability/generality of the approach to unseen attacks.

**Reviewer Concerns:**

The reviewer concerns, particularly those from Reviewers oWnt and pxv9, mainly focus on the work's limited novelty/contribution and the proposed approach's effectiveness against unseen attacks, especially in cases where the defense may not have sufficient knowledge of the attacks (in which case the attacks may not be simple to defend against). Reviewer oWnt did not respond. Reviewer pxv9 responded but was not convinced by the authors' rebuttal.

**Reviewer Scores:**

Reviewer BpQb provided a positive rating (6) and did not respond to the rebuttal. Reviewer pxv9 (with rating 2) responded and their concerns on the approach's novelty and its effectiveness on unseen attacks remained. Reviewer oWnt (with rating 4) did not respond and it seems unlikely that they would be convinced by the rebuttal and raise their rating. While the work did show that a relatively simple strategy (few-shot ICL+CoT) could be effective against known attacks, it may not be effective against unseen attack families where the defense does not have sufficient knowledge and cannot select the effective exemplars (in which case it is hard to argue that context deception attacks are simple to defend).

---

### Decision · Program_Chairs · 2026-01-26

Reject